# A bespoke mobile application for the longitudinal assessment of depression and mood during pregnancy: protocol of a feasibility study

Jose Salvador Marcano Belisario,[1] Kevin Doherty,[2] John O'Donoghue,[1] Paul Ramchandani,[3] Azeem Majeed,[4] Gavin Doherty,[2] Cecily Morrison,[1] Josip Car[1]

► Prepublication history and additional material are available. To view these files please visit the journal online (http://dx.doi.org/10.1136/10.1136/bmjopen-2016-014469).

[1]Global eHealth Unit, Department of Primary Care and Public Health, Imperial College London, London, UK
[2]School of Computer Science and Statistics, University of Dublin Trinity College, Dublin, Ireland
[3]The Centre for Mental Health, Department of Medicine, Imperial College London, London, UK
[4]Department of Primary Care, Imperial College London, London, UK

**Correspondence to**
Dr Jose Salvador Marcano Belisario;
jose.marcano-belisario10@imperial.ac.uk

## ABSTRACT

**Introduction** Depression is a common mental health disorder during pregnancy, with important consequences for mothers and their children. Despite this, it goes undiagnosed and untreated in many women attending antenatal care. Smartphones could help support the prompt identification of antenatal depression in this setting. In addition, these devices enable the implementation of ecological momentary assessment techniques, which could be used to assess how mood is experienced during pregnancy. With this study, we will assess the feasibility of using a bespoke mobile application (app) running on participants' own handsets for the longitudinal (6 months) monitoring of antenatal mood and screening of depression.

**Methods and analysis** We will use a randomised controlled study design to compare two types of assessment strategies: retrospective + momentary (consisting of the Edinburgh Postnatal Depression Scale plus five momentary and two contextual questions), and retrospective (consisting of the Edinburgh Postnatal Depression Scale only). We will assess the impact that these strategies have on participant adherence to a prespecified sampling protocol, dropout rates and timeliness of data completion. We will evaluate differences in acceptance of the technology through a short quantitative survey and open-ended questions. We will also assess the potential effect that momentary assessments could have on retrospective data. We will attempt to identify any patterns in app usage through the analysis of log data.

**Ethics and dissemination** This study has been reviewed and approved by the National Research Ethics Service Committee South East Coast—Surrey on 15 April 2016 as a notice of substantial amendment to the original submission (9 July 2015) under the Research Ethics Committee (REC) reference 15/LO/0977. This study is being sponsored by Imperial College London under the reference number 15IC2687 and has been included in the UK Clinical Research Network Study Portfolio under the Central Portfolio Management System number 19280. The findings of this study will be disseminated through academic peer-reviewed publications, poster presentations and abstracts at academic and professional conferences, discussion with peers, and social media. The findings of this study will also inform the PhD theses of JSMB and KD.

### Strengths and limitations of this study

► This study will explore (1) the role of mobile technology as a medium to address some of the practical barriers preventing depression screening in antenatal settings; (2) how mood and depression are experienced throughout pregnancy (using momentary, experiential and ecological data) and (3) two critical success factors for the successful deployment of mobile technology in pregnancy: user engagement and adherence to a proposed sampling protocol.

► This study will provide baseline information regarding the appropriateness of a sampling protocol (in terms of its duration, intensity and frequency) for the monitoring of mood and screening of depression during the antenatal period.

► The technology used in this study has been specifically designed and developed to fit within the clinical context and the local care pathways in which it will be deployed.

► This study focuses on mood and antenatal depression. It does not consider other common mental health disorders that occur during pregnancy, or the presence of potential triggers or risk factors (eg, domestic violence).

► The mood-related momentary questions used in this study have not been validated..

## INTRODUCTION

Antenatal depression is one of the most common mental health disorders during pregnancy.[1–4] Point prevalence estimates vary between 7% and 12% (depending on the trimester), and period prevalence estimates suggest that as many as 12.7% of pregnant women could experience an episode of major depression.[3] [5] Moreover, antenatal depression is associated with long-term adverse health outcomes in both mothers and their offspring. Pregnant women suffering from

depression are more likely to engage in unhealthy practices (including poor diet, substance abuse and failure to enrol in prenatal care), and are at increased risk of self-harm (or suicide) and postpartum depression.[1 6] Antenatal depression can also affect fetal development, and has been identified as an independent risk factor for a child's behavioural, cognitive and emotional development (including through adolescence).[1 7–9]

Research indicates that there is no difference in the prevalence or incidence of depression between pregnant and non-pregnant women.[10] However, the rate of diagnosis and treatment might be lower in pregnant women.[10] Approximately three-quarters of pregnant women meeting the diagnostic criteria for depression (and anxiety) are not identified, and only 1 in 10 of those who require further treatment is able to access it.[11] Some of the barriers to the prompt diagnosis of this disorder include difficulties in differentiating depressive symptoms from the expected mood and somatic changes of pregnancy, stigma, lack of reassurance that mental healthcare is a normal part of antenatal care, characteristics of healthcare providers, configuration of health services, insufficient consultation time and the cost-effectiveness of screening practices.[10–12]

As with other mental health conditions,[13–16] smartphones could help address some of the practical barriers and facilitate the screening and monitoring of depression throughout the antenatal period. The computational capabilities of these devices allow them to implement validated screening scales (usually retrospective self-reports) at any frequency and for any duration. Smartphones are also able to support the implementation of techniques for the collection of momentary, experiential and ecologically valid data. Being collected in real time, momentary data are less susceptible to many of the biases common to retrospective scales (eg, recall bias) and are more sensitive to fluctuations over time.[17–19] The networking capabilities and wide availability (approximately 71% of UK adults)[20 21] of smartphones, and the software development and distribution framework of mobile applications (apps), could help practitioners to circumvent some of the practical challenges associated with screening and clinical monitoring, and reduce the costs associated with data handling and management.

Altogether, these characteristics could contribute to making the screening and monitoring of antenatal depression more cost-effective: an initial resource-intensive app development phase would be followed by a relatively low-cost, large-scale distribution of the app onto patients who own smartphones. Thereafter, regular depression assessments could take place remotely, at anytime and anywhere (further comprehensive clinical assessments and the provision of treatment being dependent on local referral and care pathways). Nonetheless, the feasibility of using smartphones for this purpose needs to be explored.

In this area of research, a key factor is the patient's willingness to run screening or clinical monitoring apps on their personal handsets.[16] This could influence patient compliance with clinician-led data gathering (sampling) protocols, and thus affect data completeness. The latter refers to the minimum amount of information required by clinicians to inform their decisions, and is an important data quality dimension in healthcare.[21] Data completeness is also susceptible to the burden that the intensity of sampling protocols (both in terms of frequency and duration) can place on patients, as well as on the value that patients might derive from the data collected. Moreover, the impact that adding momentary assessments can have on retrospective data also needs to be explored, as this could lead to more efficient diagnostic and therapeutic decisions.

This study is part of a project aimed at understanding the role of mobile technology for the screening and assessment of antenatal depression and psychological well-being in the context of antenatal care pathways in the National Health Service (NHS) in the UK. A previous feasibility study assessed the feasibility of using iPads in the waiting area of antenatal clinics for implementing the National Institute for Health and Care Excellence (NICE) recommendations for the recognition of depression.[22] The present study will explore the feasibility of using a bespoke app to support the longitudinal and remote assessment of mood and depression screening throughout the antenatal period. We will evaluate issues of patient acceptance (namely, adherence to sampling protocols and dropout rates) by comparing two 6-month sampling protocols requiring either (1) monthly retrospective *and* momentary assessments or (2) monthly retrospective assessments.

## METHODS AND ANALYSIS
### Study design
We will assess the feasibility of using a bespoke mobile application, called BrightSelf, running on participants' own smartphones to assess and monitor depression and mood during the antenatal period through a combination of retrospective assessments and ecological momentary assessments (EMA).

We will use a parallel, randomised controlled study design to assign our participants to one of two types of assessment strategies:

1. Retrospective plus momentary assessment: requiring the completion of (1) the Edinburgh Postnatal Depression Scale (EPDS), (2) five momentary questions (assessing a participant's mood, sleep, worry, enjoyment and energy levels), and (3) two contextual questions once a month for 6 months
2. Retrospective assessment: requiring the completion of the EPDS once a month for 6 months.

### Sample selection and recruitment
We will select our sample of participants from women attending antenatal clinics in general practices, community services and secondary care NHS centres in England during their first 14 weeks of pregnancy. We have chosen

**Table 1**  Participant inclusion and exclusion criteria

| Inclusion criteria | Exclusion criteria |
| --- | --- |
| Women who are 18 years old or older | Current diagnosis of depression or other mood disorder made by a health professional |
| Up to 14 weeks pregnant (assessed through a dating ultrasound scan) | Currently receiving treatment for depression or other mood disorder (whether it is talking therapies or pharmacological treatment) |
| Any parity | Recent personal history of depression or other mood disorder in the past 12 months |
| Attending antenatal clinics in participating National Health Service sites | Not comfortable reading and writing in English |
| Own smartphone (either an iPhone or any type of Android handset) | Not owning a smartphone, or owning an incompatible handset (ie, Windows Phone, Blackberry or Linux) |

this limit to ensure that most assessments will occur during pregnancy.

On the day of their antenatal appointment, each potential participant will be approached by a clinical studies officer (CSO) or a research midwife with appropriate good clinical practice training, and will be provided with a participant information sheet. Potential participants expressing their interest in taking part in this study will be assessed against our inclusion and exclusion criteria (table 1).

Potential participants meeting our inclusion criteria will have all the study details explained to them, and will be given the opportunity to ask as many questions about the study as they need. Potential participants will have a minimum of 24 hours to decide on participation; refusal to take part in this study will not have an impact on their legal rights, medical care or their relationship with care providers.

We will obtain written informed consent from those potential participants who, after receiving all the relevant study information and having all their questions answered to their satisfaction, still wish to take part in this study.

After obtaining consent, participants will be asked to self-complete a baseline assessment using a tablet computer. This assessment will consist of a (1) sociodemographic survey, (2) the Whooley questions and (3) the EPDS (see online supplementary appendix 1). The last two instruments are recommended by the NICE to screen for depression during pregnancy.[23] Subsequently, the CSO or research midwife will guide participants through the process of downloading (from either the Apple AppStore or Google Play Store) and installing BrightSelf onto their own handsets. To this end, they will be able to use a recruiter booklet provided by the central research team (see online supplementary appendix 2). In order to activate the app, participants will need to enter a nine-digit activation code, which will also be provided by the central research team.

## Interventions to be measured

### Surveys

#### Non-validated, sociodemographic survey

We will administer an 11-question survey to collect information about participants' age group, ethnic background, marital status, employment status, level of education, smartphone and tablet computer ownership, obstetric history, and personal history of depression.

#### Whooley questions

The Whooley questions were developed as a case-finding instrument for depression in primary care.[24] This two-question instrument assesses depressed mood and anhedonia that have been present during the past month. Respondents are required to answer *Yes* or *No* to each question.

#### Edinburgh Postnatal Depression Scale

The EPDS is a 10-item self-administered survey that was originally developed to screen for postpartum depression.[25] Since then it has been validated for use in the perinatal period and for use in community and clinical settings. This instrument assesses feelings of guilt, sleep disturbance, anhedonia and suicidal ideation that have been present during the past 7 days. Each question is scored on a 4-point scale ranging from 0 to 3 points. An overall score is generated from the sum of these responses.

Although there is variability in the diagnostic accuracy of different EPDS scores, the following thresholds are commonly used: 10 points for possible depression, 13 for probable depression and 15 for antenatal depression.[26] In addition, special attention should be paid to item 10, as it deals with suicidal thoughts. Based on these scores, a clinician would be prompted to refer a woman to a mental health professional.

The EPDS is a valid and reliable tool for identifying women who are at risk of depression, both during pregnancy and postpartum. This instrument is also sensitive to changes in the severity of depression over time.[25] The EPDS can be reproduced without further permission provided that the original source of the scale is cited in each reproduced copy.

 

### Non-validated, momentary mood questions

We will administer five momentary questions to assess participants' mood, sleep, worry, enjoyment and energy (see online supplementary appendix 3). These questions are based on the work of a research fellow at the Collaboration for Leadership and Applied Health Research and Care for the East of England.[27] Each question will be mapped onto 5-point pictorial scales, ranging from 1 (low) to 5 points (high). For the purpose of this feasibility study, we will not perform any overall score calculation or attempt any validation of these questions.

### Non-validated, contextual questions

Two contextual questions will complement the momentary mood questions (see online supplementary appendix 3). They will assess (1) participants' location and (2) the activity in which they were engaged at the time they were required to complete the five momentary mood questions.

### Non-validated, poststudy acceptance survey

We will administer 13 questions (to participants completing retrospective assessments) or 14 questions (to participants completing retrospective plus momentary assessments) at the end of the 6-month participation period (see online supplementary appendix 4). The purpose of these questions is to assess the acceptability to participants of BrightSelf in the context of their antenatal care, and to gather information about their experience of using it. These questions were derived from the Usefulness, Satisfaction and Ease of Use (USE) questionnaire[28] (which focuses on usability) and also include a question concerning the desire to continue use (as used to assess engagement[29]), as well as questions regarding the experience of use and self-report. We will administer them as a web survey through Snap survey software[30] by sending participants a link via e-mail or short messaging service (SMS).

### Mobile system

BrightSelf is a mobile system for the collection of self-reports, both retrospective and momentary, during pregnancy. This system is not a diagnostic tool and is not intended to replace the role of clinicians within antenatal or mental healthcare pathways.

This system elicits retrospective reports through the EPDS, and momentary reports according to the five constructs of mood, sleep, worry, enjoyment and energy. In addition, the system can suggest brief activities that users can perform to lift their mood, additional information about the system and the feasibility study, additional resources should they need immediate help, and a visualisation of past self-reports. We will emphasise to the participants that BrightSelf is not a clinical tool, and that if they feel they need immediate help at any point during the study they should contact their general practitioners.

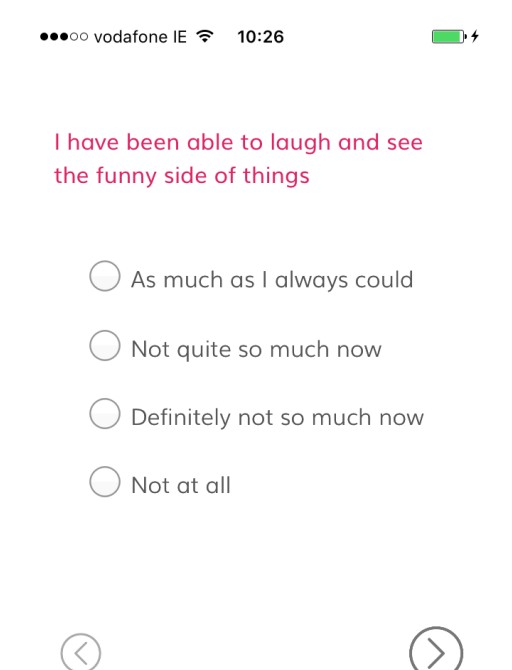

**Figure 1** BrightSelf—Check Back (screenshot).

BrightSelf consists of mobile applications for Android and iOS operating systems, a backend for the storage and management of data, and a website that supports the monitoring of data.

A complete description of BrightSelf is the aim of another publication. Here we describe the two features that are most relevant to this feasibility study, namely the collection of retrospective and momentary self-reports.

### Check Back

This feature will enable the administration of the EPDS. It consists of two introductory screens informing participants of the retrospective nature of this scale and its intended applications. These screens will be followed by the 10 EPDS questions, presented one question per screen, using radio buttons to capture participants' responses (figure 1).

### Check In

This feature will enable the administration of the five momentary questions and the two contextual questions. The first five questions will be presented using a 5-point pictorial scale with temporary supporting text (see figure 2 for an example). The last two questions will use radio buttons to capture participants' responses.

### Retrospective versus momentary assessments

Participants will be asked to complete monthly assessments at irregular intervals for 6 months. Whenever an assessment is due (as per the sampling protocol), a notification will be displayed on a participant's handset. There will be two types of notifications: one for the *Check Back* feature and one for the *Check In* feature of BrightSelf. The visual appearance of the notifications will depend on the operating system and on the model of the participant's handset. The text of the

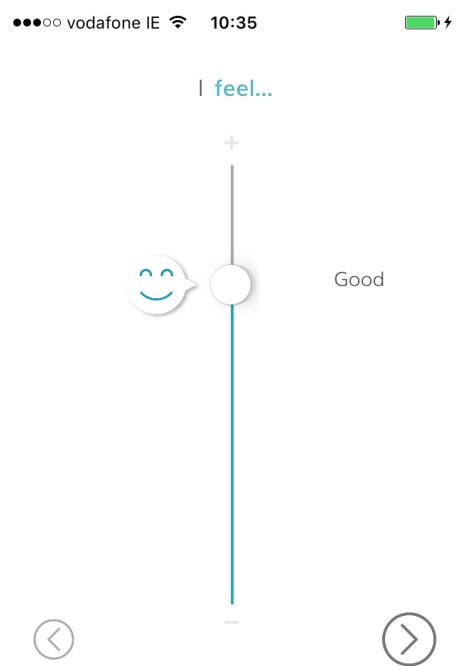

**Figure 2** BrightSelf—Check In (screenshot).

notifications, however, will remain constant across participants for each type of notification.

Participants will be able to respond (and thus complete the corresponding assessments) or to dismiss the notifications. In addition, participants will be able to leave notifications unanswered. If participants do not complete an assessment in response to a notification, they will not receive reminders or follow-up notifications. In these instances, the non-response will be recorded in our database and will be coded as such for the purposes of our data analysis.

Outside assessment periods, participants will be able to use BrightSelf in any way they find convenient and there will be no restrictions on the number of assessments they can complete on a single day. However, we will be able to distinguish between assessments completed in response to a notification and those completed spontaneously. At baseline, participants will be randomly allocated to one of two types of assessment strategies:

### Retrospective plus momentary assessments

In this experimental manipulation, participants will be asked to complete a combination of retrospective assessments and momentary assessments. A retrospective assessment will be defined as a single administration of the EPDS. A momentary assessment will be defined as the five momentary questions plus the two contextual questions.

Participants will be required to complete assessments for a total of 6 months, and each assessment period will consist of six consecutive days. The 6-month participation period will be calculated from the day on which the app is activated (ie, when participants enter their activation code and activate their account), and it will be divided into six intervals. All participants will receive

the first notification 2 days after activating BrightSelf (ie, first assessment), while the use of the app is fresh within their minds and to give an idea of what to expect during the study. Subsequent assessment periods will be a random selection of six consecutive days within the 21–35 days following the end of the previous assessment period (in order to avoid 12 days of continuous assessments).

The assessment period will be structured as follows:
► Day 1: one retrospective assessment at any random time between 17:00 and 21:00
► Days 2–5: three momentary assessments per day, displayed at random times within each of the following intervals: 09:00–12:00, 13:00–16:00 and 17:00–20:00
► Day 6: one retrospective assessment at any random time between 17:00 and 21:00.

At the end of the 6-month period, participants will be sent a link via SMS or by e-mail to complete the non-validated, poststudy acceptance survey.

### Retrospective assessments

In this experimental manipulation, participants will be asked to complete retrospective assessments only. A retrospective assessment will be defined as a single administration of the EPDS.

Participants will be required to complete assessments for a total of 6 months, and each assessment period will consist of 1 day. The 6-month participation period will be calculated from the day on which the app is activated (ie, when participants enter their activation code and activate their account), and it will be divided into six intervals. Again, all participants will receive the first notification 2 days after activating BrightSelf (ie, first assessment). Subsequent assessments will take place on a random day within the 21–35 days following the end of the previous assessment period (in order to avoid two consecutive EPDS assessments).

On each assessment day, participants will receive a notification to complete the EPDS at a random time between 17:00 and 21:00.

At the end of the 6-month period, participants will be sent a link via SMS or by e-mail to complete the non-validated, poststudy acceptance survey (see figure 3 for an illustration of both sampling protocols).

### Duty of care

Regardless of participants' allocation, the central research team will be alerted if any of a series of prespecified conditions are met (table 2). These alerts will be colour-coded according to their severity level. In order to determine the severity level, we have chosen commonly used EPDS scoring thresholds (ie, 10–12 points for possible depression, and 13 points or more for probable depression). We acknowledge the limitations of this approach as the diagnostic accuracy of the EPDS can vary depending on the setting and the population in which it is administered.

**Figure 3** Sampling protocols for the retrospective plus momentary assessments and for the retrospective assessments. EPDS, Edinburgh Postnatal Depression Scale.

Before starting the study, the central research team will agree on a list of designated contacts with the relevant clinical care teams. These contacts will include clinicians who are available during normal working hours, as well as those on duty outside working hours (including weekends and bank holidays). If a red or orange alert is generated, the study coordinator will contact the designated member of the clinical care team by phone and e-mail within 24 hours of receiving the alert. The clinical care team will then follow up these alerts directly with the participants.

### Randomisation

We will use block randomisation procedures (with blocks of four) to allocate participants to one of the two experimental arms. Random numbers will be generated using Stata V.14.[31] Each consecutive number will be embedded within the nine-digit activation codes that will be distributed to each participating NHS site. The full activation codes will be generated in the same order in which the random numbers were generated. Each NHS site will receive a list of activation codes on a first-come-first-served basis. Members of the local research teams will not be informed of the random sequence generation and will not be able to identify which number determines participant allocation. They will be required to use activation codes sequentially, as participants are recruited. Activation codes referring to the *retrospective plus momentary assessment* condition will activate a version of BrightSelf in which both the *Check Back* and the *Check In* features are active. Activation codes referring to the *retrospective assessment* condition will activate a version of BrightSelf in which only the *Check Back* feature is active.

### Outcomes

#### Adherence to sampling protocols

We will calculate the number of participants who complete 100% of the expected assessments (as per the study protocol) as a proportion of the total number of participants who were randomised into the study. We will subdivide this outcome by using the total number of participants who complete the 6-month participation period as the denominator.

| Type of alert | Criteria |
|---|---|
| **Table 2** Criteria for an alert to be sent to the research team | |
| Yellow (Mild) | Completing more than one EPDS assessment on the same day, with an overall EPDS score of 9 points or less and a score of 0 on question 10 of the EPDS |
| Orange (Moderate) | Overall EPDS score between 10 and 12 points, with a score of 0 on question 10 of the EPDS |
| Red (Severe) | Overall EPDS score of 13 points or more, and 1 or more points on question 10 of the EPDS, regardless of the overall EPDS score |

EPDS, Edinburgh Postnatal Depression Scale.

## Dropout rates

We will calculate the number of participants who complete the 6-month participation period as a proportion of the total number of participants who were randomised into the study.

## Usage patterns

We will assess participants' usage pattern of BrightSelf by analysing log data (additional variables capturing app usage). These will include the number of additional voluntary self-reports, time spent using the app, number of interactions with other sections of the app, time taken to complete the self-assessments and other ancillary usage data.

## Acceptance

We will calculate participants' ratings and responses to the poststudy acceptance survey. In addition, we will conduct thematic analysis of participants' answers to the open-ended questions of this survey using Atlas.ti 8.[32]

## Timeliness of data completion

We will calculate the number of assessments that were completed in response to a notification as a proportion of the total number of expected assessments. We will subdivide this outcome by using the total number of completed assessments as the denominator. We will consider that an assessment has been completed in response to a notification if it takes place within the interval corresponding to that notification (ie, before the next notification is delivered). This broad interval may raise concerns regarding the ecological validity of the report, which we will explore in our statistical analysis.

## Sample size calculations

We have chosen to relate the proposed sample size to the 95% CI for the adherence rate, as recommended by the Research Design Service in London. Therefore, we would need 96 participants in each arm with a 95% confidence level and a CI of 10. This translates into a total sample of 192 (ie, 96 participants in each experimental group). For this reason, we will aim to recruit at least 250 participants to account for dropouts and potential miscarriages.

## Data analysis plan

### Descriptive statistics

We will report the number of potential participants who were eligible and refused to take part in the study. Where possible, we will report the reasons for refusing to participate.

For each experimental group, we will report the following information:

► Demographic characteristics (as captured by the non-validated, sociodemographic survey)
► Proportion of participants answering Yes to any of the Whooley questions during the baseline assessment
► Proportion of participants scoring at each interval of the EPDS: between 0 and 9 points, between 10 and 12 points, and 13 points or above during the baseline assessment

► Proportion of participants scoring 1 or more points on question 10 of the EPDS during the baseline assessment.

### Inferential statistics

We will compare the *retrospective and momentary assessment* and the *retrospective assessment* experimental groups for differences in adherence rates, dropout rates and timeliness of data completion. For this we will use a t-test or the non-parametric equivalent.

We will compare acceptance between the two experimental manipulations by assessing differences in participants' rankings to those questions on a 7-point Likert scale of the poststudy acceptance survey. We will also conduct a thematic analysis of the open-ended questions of this survey.

In relation to the momentary assessments, we will examine the distribution of delays between the time of notification and time of report in order to assess the effect of this delay on the ecological validity of reports.

We will analyse usage patterns through regression modelling of log data. We will analyse participants' momentary and retrospective assessments through time-based analyses or multilevel modelling.[33 34] We will attempt to compare if the momentary assessments had any effect on the retrospective assessments by comparing the EPDS responses given on day 1 to those given on day 6.

## Timeline

We expect participant recruitment to start in January 2017, and the last follow-up to take place in August/September 2017 (assuming a recruitment period of 2 months and a half).

## CONCLUSION

This study addresses an important area of unmet clinical need, with direct and indirect consequences for mothers, children, their families, health systems and society. This study will contribute to the growing body of evidence concerning the role of mobile technologies for the support of mental health. Similarly, it will generate baseline information concerning the acceptability of an EMA sampling protocol (in term of its duration, frequency and intensity) to women attending antenatal care. This study will also evaluate participant engagement with the technology and their adherence to a prespecified sampling protocol, both of which influence the completeness of the data needed by clinicians to inform their decisions. In addition, this study will identify some of the implementation issues that might arise when we attempt to deploy mobile technologies in clinical settings, which could affect their successful adoption.

In this work, we have maintained a focus on antenatal depression. Mental health during pregnancy, however, is more complex. Disorders such as anxiety and post-traumatic stress disorder are also among the most common disorders during this period. Moreover, pregnant women who are exposed to certain risk factors or triggers (eg,

domestic violence, social isolation, minority groups and low income) are at increased risk of suffering from any of these mental health problems. Although we have not been able to focus on all these issues, we believe that the findings from this feasibility study will produce important lessons that could enable us to design and develop similar technologies and appropriate strategies. We aim to recruit participants from diverse backgrounds, and across multiple settings and geographical areas within England, and to focus on how such technologies can be embedded within existing antenatal care pathways, in order to support the existing patient–midwife relationship.

The findings from this and from a previous feasibility study[21] will inform a larger trial evaluating the integration of mobile technologies into routine antenatal pathways, and their potential impact on clinical outcomes.

**Acknowledgements** We would like to thank Professors Susan Ayers and Alexandra Thornton of City University London, Joanna Girling and Marie O'Connell of West Middlesex Hospital, Tara Pauley and Charlotte Clayton of Hinchingbrooke Hospital, Antoinette McNulty and Gemma Loebenberg of the Northwest London CRN, and Dr Fiona Blake for their feedback on early versions of BrightSelf and the study protocol, and for their help with our patient and public involvement initiatives. We would also like to thank Marguerite Barry of Trinity College Dublin for running some of the patient and public involvement sessions with us.

**Contributors** JSMB developed the study protocol in collaboration with KD. JSMB was in charge of obtaining ethics and governance approvals and has been approaching potential participant recruitment centres. KD was in charge of software development. JSMB drafted this manuscript. CM, GD, PR, JOD, AM and JC guided the development of the study protocol. CM, JOD and JC have supervised JSMB's work. GD has supervised KD's work. All the authors reviewed and approved this manuscript.

**Funding** This work is being supported by a National Institute for Health Research (NIHR) Imperial Biomedical Research Centre (BRC) award through the Population Health Theme, and by the ADAPT Centre through Trinity College Dublin.

**Competing interests** None declared.

**Ethics approval** Research Ethics Committee South East Coast – Surrey.

**Provenance and peer review** Not commissioned; externally peer reviewed.

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
