## [Reviewer comments · BMJ Open]

ARTICLE DETAILS

TITLE (PROVISIONAL)	A bespoke mobile application for the longitudinal assessment of depression and mood during pregnancy: protocol of a feasibility study
AUTHORS	Marcano Belisario, Jose Salvador; Doherty, Kevin; O'Donoghue, John; Ramchandani, Paul; Majeed, Azeem; Doherty, Gavin; Morrison, Cecily; Car, Josip

VERSION 1 - REVIEW

REVIEWER	Louise Howard Kings College London UK
REVIEW RETURNED	16-Oct-2016

GENERAL COMMENTS	Abstract- could you include the way acceptance of the technology is being measured in the Methods section ie short quantitative survey and open ended questions. The last sentence of the final limitation listed is rather vague and could be clarified - what challenges are being referred to and how could this technology circumvent which disadvantages of self report measures? Main manuscript: Background: reference is made to cost-effectiveness and one reference cited but this paper has been superceded by more recent analyses (with differing findings) including in recent NICE update and in more recent evidence syntheses (eg Lancet. Howard et al 2014 - apologies for self citing here). Method: could the authors clarify the mode of administration for the baseline assessment ie will the women be asked the whooley questions for example or self complete it using paper or a mobile device? (It could be interesting to compare researcher administered baseline whooley responses with real life routine practice responses to clinical midwives? Inclusion criteria: how will the research team know whether the woman has booked in the first trimester of pregnancy and not later? Why exclude women booking later? Clinical teams will be notified if responses indicate amber or red. Do the women know this is the case and could this bias their responses? Does this mean midwives will be notified and if so how? What happens if a midwife is on leave? What safety procedures are in place? The Brightself welcome guide states that if responses indicate a woman is struggling the research team will be notified. I
--

	think this may be a typo and will need to be altered to clarify to women the process and who of their clinicians will be informed. It may be prudent to state that it will take some time for the clinical team to make contact potentially and that if a woman is feeling she needs help urgently then she should contact her GP. It may be difficult to get midwives (or GPs if this is planned) to see women quickly if they are informed of a red response. How were the post study acceptance survey questions developed and piloted? Some of the questions are open ended and require comments - how will this be collected and analysed as some form of qualitative data management tool and analysis will be needed. Please add details. Could the authors provide appendices of the extra materials that will be included in the Brightself app?
--	--

REVIEWER	Prof. Dr. VJM Pop Department of Medical psychology Tilburg University The Netherlands
REVIEW RETURNED	18-Nov-2016

GENERAL COMMENTS	First of all, I want to apologize for my perhaps "direct" English, it is not personally meant. I am not a native speaker so I definitely will miss the nuances of UK English General comments It is always interesting when one has to review papers or research proposals in your own (research experience of over 25 years) topic. Especially when one finds out rapidly that the group who presents in this case their proposal has no real expertise on this topic. Looking at the group, there seems to be a broad experience in computer science and the public health domain while any expertise in the field of perinatal mental health is lacking. I would strongly advise – knowing of course the very prestigious level of the Imperial College of London – the research group to look for collaborators with a perinatal research groups, for example the famous Maudsley Perinatal Psychiatry Clinic of London or the Oxford group of Murray and Cooper. The lack of real experience is generally reflected by statements that are not totally in line with up to date knowledge in this particular area of research, which is in my case, is perinatal mental health (I am not a computer scientist). This can be regarded as a minor problem (I will come with examples later on) but it becomes more difficult when the authors are really "missing the point". Because the basic instrument to assess perinatal depressive symptoms of their proposal is the Edinburgh (Postnatal) Depression Scale (EPDS or EDS), developed by John Cox (UK) back in the eighties, it is rather strange that they give information which is false. On page 9 the authors write: "this (EPDS) instrument assesses feelings of guilt, sleep disturbance, reduced energy levels, anhedonia and suicidal ideation that have been present during the past 7 days". If one looks in the appendix where the EPDS is shown, one can already see that there is no item referring to "reduced energy levels". This seems to be a minor slip of the pen but for me the authors show that they do not understand why and how the EPDS has been developed.
---

Because low energy levels is of little if any value during pregnancy or the postpartum period (every pregnant women or those having delivered a baby in the last 6 months will report low energy levels) John Cox intentionally omitted this - and other somatic items which are less relevant to the postpartum period such as loosing / gaining weight, decreased / increased appetite. (Interestingly Michael O'Hara's group of the US showed many years later that the properties of the EPDS do not become worse when somatic items commonly seen in depression are not taken out of the questionnaire).

A second important comment is the use of the Whooley questions. If one looks at the reference list with the tempting title: " Case-finding instruments for depression – two questions are as good as many", I would like to add: few reliable instruments are as good as many. The PHQ-2 – or the PHQ-4 if one would like to add general anxiety items – is a very reliable instrument which has shown to have excellent properties in picking-up people at risk for syndromal depression in numerous papers published the last decades. It is depicted from the PHQ-7 (patient health questionnaire 7 items) and also covers the two main symptoms of depression: anhedonia and depressed mood. However, it gives the opportunity to answer on a four point Likert scale, in line with the EPDS.

Another comment is from a clinical point of view. Being involved myself in treatments of depression / anxiety by internet, I see the advantages of computer science in medicine. However, it should be realized that – at least in The Netherlands and presumably also in UK – a pregnant woman sees an obstetric health care worker (nurse, midwife, obstetrician) at least 10 times during normal pregnancy. Instead of offering computer based diagnostics to a (sub: 70%)group of pregnant women, a more logic alternative to me is to introduce simple questionnaires to the standard obstetric consultation to all pregnant women to pick-up women at risk for depression and I know that this is happening already for long time in UK. In other words, I do not see the additive value of computer diagnostics during pregnancy. Of course the authors are right that during the postpartum period the regular consultations have disappeared. This period seems to be an attractive period for computer based diagnostics. However, in the summary they state that the current proposal especially focus on pregnancy. Having said this, it is well known from perinatal research that the risk factors for depression during pregnancy are almost identical to those during the postpartum period.....

A final comment is the nuance that depression is a common disorder during pregnancy and postpartum. Yes, this is true, but depression in the perinatal period is no different in prevalence and characteristics compared to non-childbearing women of the same age. This goes back to Lewinsohn's work showing that the two most important determinants of depression are: being young and being female.

Specific comments

Page 4: strengths of the study: "this study explores the role of mobile technology as a medium to (i) address some of the barriers preventing depression screening in antenatal settings", barriers of patients or health care workers??? see my previous comments.

Same page: "this study will provide baseline information regarding the appropriateness of a sampling protocol (in terms of its duration, intensity and frequency)

for the monitoring of mood and depression, I would say: mood problems.

Page 5:

INTRODUCTION

"Antenatal depression is one of the most common, treatable mental health disorders in pregnancy.[1-4]"

Up to 2% of pregnant women already suffer from chronic depression in which treatment fails to show little if any benefit.

Line 42 page 5 to line 9 of page 6: all the advantages of smartphone use can easily be overruled by the simple implementation of standard mental health diagnostics during the regular obstetric consultation. Moreover, the clinical view of a midwife has tremendous advantage over self-rating reports (be it by smartphone or paper and pencil). Therefore I really doubt the cost-effectiveness of smartphone use during pregnancy (not the postpartum period). During consultation there is no risk of retrospective information: the client sits in front of the health care worker.

Page 7: "Retrospective assessment: requiring the completion of the EPDS at irregular intervals once a month for 6 months," irregular intervals once a month for 6 months seems very regular to me.

Page 7-8: exclusion criteria: up to 10% of the pregnant women in The Netherlands use benzodiazepines / antidepressant drugs. Are they excluded in the current study when a woman has no depression, what means the word currently being treated?

Same page: "Potential participants will have at least 24 hours to decide on participation" being a former member of a Medical Ethical Review Board, the minimum of one week was commonly requested when recruiting participants.

Same page: "We will administer an 11-question survey to collect information about participants' age group, ethnic background, marital status, employment status, level of education, smartphone and tablet computer ownership, obstetric history, and previous personal history of depression."

Please give details: for example, unplanned pregnancy is an important determinant of pregnancy depression. Complications during previous pregnancies / deliveries.

Page 9: " The EPDS is a 10-item self-administered survey that was developed to screen for perinatal depression in the community.[24]"

No, it was originally developed to screen for postpartum depression.

Same page: "scores of 13 points or more suggest that the diagnostic criteria for major

depression disorder have probably been met.[25]"

Our group was among the first to validate the EPDS during pregnancy. With acceptable sensitivity / specificity, the PPV of a score of 13 or higher will never be higher than 50-60%. What do the authors mean by "probably"?

I am aware of a large (thousands of participants) multi-center meta-analysis which is currently under way to question the reliability of the EPDS as an appropriate screening instrument for depression in the perinatal period.

Same page: " We will administer 5 momentary questions to assess participants' mood, sleep, worry, enjoyment and energy (Appendix 3). These questions are based on the work of a research fellow at the Collaboration for Leadership and Applied Health Research and Care (CLAHRC) for the East of England.[26] Each question will be mapped onto 5-point pictorial scales, ranging from 1 (low) to 5 points (high). For the purpose of this feasibility study, we will not perform any overall score calculation or attempt any validation of these questions."

So then, what is the relevance of adding these questions if it is not for validation?

Strictly speaking: worrying is a major symptom of anxiety which is not a mood symptom. Once again, the low energy question will not discriminate at all during pregnancy.

Page 12 " In this experimental manipulation, participants will be asked to complete a combination of retrospective assessments and momentary assessments. A retrospective assessment will be defined as a single administration of the EPDS. A momentary assessment will be defined as the 5 momentary questions plus the 2 contextual questions."

To me this is a major flaw of the design: the authors use a non-validated instrument (5 momentary questions) to discriminate between two arms of a protocol.

Same page: " The assessment period will be structured as follows:

- Day 1: one retrospective assessment at any random time between 17:00 - 21:00;
- Day 2 to 5: 3 momentary assessments per day, displayed at random times within each of the following intervals: 09:00 – 12:00; 13:00 – 16:00; and 17:00 – 20:00; and
- Day 6: one retrospective assessment at any random time between 17:00 and 21:00."

The latter retrospective refers to the previous 7 days?

Page 15: " We have chosen to relate the proposed sample size to the 95% confidence interval for the adherence rate, as recommended by the Research Design Service in London.

Therefore, we would need 96 participants in each arm with a 95% confidence level and a confidence interval of 10. This translates into a total sample of 192 (i.e., 96 participants in each experimental group). For this reason, we will aim to recruit at least 200 participants, to account for drop-outs."

Because the authors start very early during pregnancy do they incorporate abortion drop-out, which can be substantially until 16 weeks?

Same page: " We will compare the Retrospective and Momentary assessment and the Retrospective assessment experimental groups for differences in adherence rates, drop-out rates and timeliness of data completion."

To be honest, until this paragraph, the rationale for the comparison of these two approaches was not clear to me and still is not very clear. Normally in trials, there is a (null) hypothesis: what do the authors expect and based on which assumptions?

Page 53: The rationale of the contextual questions is not clear to

	me. What do the authors think to conclude from the answers given by the respondents?
--	--

VERSION 1 – AUTHOR RESPONSE

REVIEWER 1:

ABSTRACT- COULD YOU INCLUDE THE WAY ACCEPTANCE OF THE TECHNOLOGY IS BEING MEASURED IN THE METHODS SECTION IE SHORT QUANTITATIVE SURVEY AND OPEN ENDED QUESTIONS.

We have indicated in the abstract that user acceptance will be measured using a short quantitative survey and open ended questions.

THE LAST SENTENCE OF THE FINAL LIMITATION LISTED IS RATHER VAGUE AND COULD BE CLARIFIED - WHAT CHALLENGES ARE BEING REFERRED TO AND HOW COULD THIS TECHNOLOGY CIRCUMVENT WHICH DISADVANTAGES OF SELF REPORT MEASURES?

We have decided to remove this last bullet point, as we feel it is outside the scope of the work outlined by this study protocol.

BACKGROUND: REFERENCE IS MADE TO COST-EFFECTIVENESS AND ONE REFERENCE CITED BUT THIS PAPER HAS BEEN SUPERCEDED BY MORE RECENT ANALYSES (WITH DIFFERING FINDINGS) INCLUDING IN RECENT NICE UPDATE AND IN MORE RECENT EVIDENCE SYNTHESSES (EG LANCET. HOWARD ET AL 2014 - APOLOGIES FOR SELF CITING HERE).

Thank you for bringing this to our attention. We have updated the references cited to include: Howard LM, Molyneaux E, Dennis CL, et al. Non-psychotic mental disorders in the perinatal period. Lancet 2014;384(9956):1775-88 doi: 10.1016/S0140-6736(14)61276-9 [published Online First: 14 November 2014].

METHOD: COULD THE AUTHORS CLARIFY THE MODE OF ADMINISTRATION FOR THE BASELINE ASSESSMENT IE WILL THE WOMEN BE ASKED THE WHOOLEY QUESTIONS FOR EXAMPLE OR SELF COMPLETE IT USING PAPER OR A MOBILE DEVICE? (IT COULD BE INTERESTING TO COMPARE RESEARCHER ADMINISTERED BASELINE WHOOLEY RESPONSES WITH REAL LIFE ROUTINE PRACTICE RESPONSES TO CLINICAL MIDWIVES?

We have clarified that participants will be asked to self-complete the baseline assessment using a tablet computer [page 9, paragraph 3 of the revised manuscript]. We agree with your suggestion that it would be interesting to compare research-administered baseline Whooley responses with midwife-administered Whooley responses. However, participants will be recruited outside the clinical consultation, which will prevent us from being able to make this comparison.

INCLUSION CRITERIA: HOW WILL THE RESEARCH TEAM KNOW WHETHER THE WOMAN HAS BOOKED IN THE FIRST TRIMESTER OF PREGNANCY AND NOT LATER? WHY EXCLUDE WOMEN BOOKING LATER?

We have amended our inclusion criteria table and methods sections to indicate that we will include participants who are up to 14 weeks pregnant, as indicated by a dating scan [pp. 8 and 9 of the revised manuscript, including Table 1]. The reason for this limit is to ensure that most of the follow-up assessments fall within pregnancy.

CLINICAL TEAMS WILL BE NOTIFIED IF RESPONSES INDICATE AMBER OR RED. DO THE WOMEN KNOW THIS IS THE CASE AND COULD THIS BIAS THEIR RESPONSES? DOES THIS MEAN MIDWIVES WILL BE NOTIFIED AND IF SO HOW? WHAT HAPPENS IF A MIDWIFE IS ON LEAVE? WHAT SAFETY PROCEDURES ARE IN PLACE? THE BRIGHTSELF WELCOME GUIDE STATES THAT IF RESPONSES INDICATE A WOMAN IS STRUGGLING THE RESEARCH TEAM WILL BE NOTIFIED. I THINK THIS MAY BE A TYPO AND WILL NEED TO BE ALTERED TO CLARIFY TO WOMEN THE PROCESS AND WHO OF THEIR CLINICIANS WILL BE INFORMED. IT MAY BE PRUDENT TO STATE THAT IT WILL TAKE SOME TIME FOR THE CLINICAL TEAM TO MAKE CONTACT POTENTIALLY AND THAT IF A WOMAN IS FEELING SHE NEEDS HELP URGENTLY THEN SHE SHOULD CONTACT HER GP. IT MAY BE DIFFICULT TO GET MIDWIVES (OR GPs IF THIS IS PLANNED) TO SEE WOMEN QUICKLY IF THEY ARE INFORMED OF A RED RESPONSE.

Participants will be made aware that their EPDS scores might generate an alert, in which case the clinical care teams will be notified (this is included in the informed consent form).

We have added a new sub-section in the Methods section outlining our safety procedures in case a red/orange alert is generated [pp. 14 & 15, labelled Duty of Care]. Overall, before commencing recruitment the central research team will agree with the recruitment sites a list of designated clinicians that can be contacted should an alert be generated. This list will include clinicians who are available during working hours, as well as those who will be available outside working hours (including weekends and bank holidays). Upon receiving an alert, the study coordinator will contact the designated member of the clinical care team by phone and email within 24 hours.

Although this phenomenon has been reported for the EPDS when administered in clinical settings, we don't know if it will bias participants' responses in this study. Remote administration of this instrument could confer a certain level of anonymity, that encourages participants to answer more truthfully. Certainly, this is something to consider when assessing results.

Regarding the welcome guide, this is not intended for participants to take home with them. Instead, it is aimed as a guide for research midwives or CSOs to go through the process of downloading and installing BrightSelf with the participants in the clinic. In addition, the SOPs provided by the central research team asks research midwives and/or CSOs to emphasise that BrightSelf is not intended to replace clinicians, and that if participants feel they need help they should contact their GP. This information is also included in BrightSelf. Within BrightSelf, we also provide a list of useful contacts (for example, Samaritans and Mind) should participants need additional support.

HOW WERE THE POST STUDY ACCEPTANCE SURVEY QUESTIONS DEVELOPED AND PILOTTED? SOME OF THE QUESTIONS ARE OPEN ENDED AND REQUIRE COMMENTS - HOW WILL THIS BE COLLECTED AND ANALYSED AS SOME FORM OF QUALITATIVE DATA MANAGEMENT TOOL AND ANALYSIS WILL BE NEEDED. PLEASE ADD DETAILS.

The post-study surveys were derived from the USE questionnaire (which focuses on usability – although this is a validated scale, we borrowed some questions from it) and also include a question concerning the desire to continue use (as used to assess engagement), as well as questions regarding the experience of use and self-report [changes made on page 11 under the sub-heading 'Non-validated, post-study acceptance survey']. The relevant references have also been added to the manuscript:

- Lund AM. Measuring usability with the USE questionnaire. *Usability Interface* 2001;8(2):3-6
- Schoenau-Fog H. (2011). Hooked! – evaluating engagement as continuation desire in interactive narratives. In *Interactive storytelling* (pp. 219 – 230). Springer: Berlin, Heidelberg

Although our post-study survey is not validated, it measures domains relevant to user engagement in the human computer interaction literature.

We will administer these questions as a web survey using Snap® surveys. After the 6-month follow up period, we will send participants a link to this survey via email or SMS. In relation to participants' answers to the open-ended questions, we will conduct thematic analysis using Atlas.ti.

COULD THE AUTHORS PROVIDE APPENDICES OF THE EXTRA MATERIALS THAT WILL BE INCLUDED IN THE BRIGHTSELF APP?

We have included a brief description of the overall functionality and content of BrightSelf [pp 11 & 12, under the sub-heading 'Mobile System']. A more detailed description of the app is the aim of a separate publication – its features broken down and detailed rationale for the inclusion of each component.

REVIEWER 2:

IT IS ALWAYS INTERESTING WHEN ONE HAS TO REVIEW PAPERS OR RESEARCH PROPOSALS IN YOUR OWN (RESEARCH EXPERIENCE OF OVER 25 YEARS) TOPIC. ESPECIALLY WHEN ONE FINDS OUT RAPIDLY THAT THE GROUP WHO PRESENTS IN THIS CASE THEIR PROPOSAL HAS NO REAL EXPERTISE ON THIS TOPIC. LOOKING AT THE GROUP, THERE SEEMS TO BE A BROAD EXPERIENCE IN COMPUTER SCIENCE AND THE PUBLIC HEALTH DOMAIN WHILE ANY EXPERTISE IN THE FIELD OF PERINATAL MENTAL HEALTH IS LACKING. I WOULD STRONGLY ADVISE – KNOWING OF COURSE THE VERY PRESTIGIOUS LEVEL OF THE IMPERIAL COLLEGE OF LONDON – THE RESEARCH GROUP TO LOOK FOR COLLABORATORS WITH A PERINATAL RESEARCH GROUPS, FOR EXAMPLE THE FAMOUS MAUDSLEY PERINATAL PSYCHIATRY CLINIC OF LONDON OR THE OXFORD GROUP OF MURRAY AND COOPER.

THE LACK OF REAL EXPERIENCE IS GENERALLY REFLECTED BY STATEMENTS THAT ARE NOT TOTALLY IN LINE WITH UP TO DATE KNOWLEDGE IN THIS PARTICULAR AREA OF RESEARCH, WHICH IS IN MY CASE, IS PERINATAL MENTAL HEALTH (I AM NOT A COMPUTER SCIENTIST). THIS CAN BE REGARDED AS A MINOR PROBLEM (I WILL COME WITH EXAMPLES LATER ON) BUT IT BECOMES MORE DIFFICULT WHEN THE AUTHORS ARE REALLY "MISSING THE POINT".

One of our team (Prof Paul Ramchandani) has extensive research experience with perinatal populations, including clinical trials. He has worked closely with the Murray/Cooper/Stein group for the past 15 years.

BECAUSE THE BASIC INSTRUMENT TO ASSESS PERINATAL DEPRESSIVE SYMPTOMS OF THEIR PROPOSAL IS THE EDINBURGH (POSTNATAL) DEPRESSION SCALE (EPDS OR EDS), DEVELOPED BY JOHN COX (UK) BACK IN THE EIGHTIES, IT IS RATHER STRANGE THAT THEY GIVE INFORMATION WHICH IS FALSE. ON PAGE 9 THE AUTHORS WRITE: "THIS (EPDS) INSTRUMENT ASSESSES FEELINGS OF GUILT, SLEEP DISTURBANCE, REDUCED ENERGY LEVELS, ANHEDONIA AND SUICIDAL IDEATION THAT HAVE BEEN PRESENT DURING THE PAST 7 DAYS". IF ONE LOOKS IN THE APPENDIX WHERE THE EPDS IS SHOWN, ONE CAN ALREADY SEE THAT THERE IS NO ITEM REFERRING TO "REDUCED ENERGY LEVELS". THIS SEEMS TO BE A MINOR SLIP OF THE PEN BUT FOR ME THE AUTHORS SHOW THAT THEY DO NOT UNDERSTAND WHY AND HOW THE EPDS HAS BEEN DEVELOPED. BECAUSE LOW ENERGY LEVELS IS OF LITTLE IF ANY VALUE DURING PREGNANCY OR THE POSTPARTUM PERIOD (EVERY PREGNANT WOMEN OR THOSE HAVING DELIVERED A BABY IN THE LAST 6

MONTHS WILL REPORT LOW ENERGY LEVELS) JOHN COX INTENTIONALLY OMITTED THIS - AND OTHER SOMATIC ITEMS WHICH ARE LESS RELEVANT TO THE POSTPARTUM PERIOD SUCH AS LOOSING / GAINING WEIGHT, DECREASED / INCREASED APPETITE. (INTERESTINGLY MICHAEL O'HARA'S GROUP OF THE US SHOWED MANY YEARS LATER THAT THE PROPERTIES OF THE EPDS DO NOT BECOME WORSE WHEN SOMATIC ITEMS COMMONLY SEEN IN DEPRESSION ARE NOT TAKEN OUT OF THE QUESTIONNAIRE).

We thank the reviewer for bringing this oversight to our attention. We have now removed reduced energy levels from the list on page 9 [now page 10 in the revised version of the manuscript].

A SECOND IMPORTANT COMMENT IS THE USE OF THE WHOOLEY QUESTIONS.

The choice of the Whooley questions is based on the recommendations of the National Institute for Health and Care Excellence in their guideline Antenatal and Postnatal Mental Health. We have added this information to the Methods section, and the corresponding reference.

ANOTHER COMMENT IS FROM A CLINICAL POINT OF VIEW. BEING INVOLVED MYSELF IN TREATMENTS OF DEPRESSION / ANXIETY BY INTERNET, I SEE THE ADVANTAGES OF COMPUTER SCIENCE IN MEDICINE. HOWEVER, IT SHOULD BE REALIZED THAT – AT LEAST IN THE NETHERLANDS AND PRESUMABLY ALSO IN UK – A PREGNANT WOMAN SEES AN OBSTETRIC HEALTH CARE WORKER (NURSE, MIDWIFE, OBSTETRICIAN) AT LEAST 10 TIMES DURING NORMAL PREGNANCY. INSTEAD OF OFFERING COMPUTER BASED DIAGNOSTICS TO A (SUB: 70%)GROUP OF PREGNANT WOMEN, A MORE LOGIC ALTERNATIVE TO ME IS TO INTRODUCE SIMPLE QUESTIONNAIRES TO THE STANDARD OBSTETRIC CONSULTATION TO ALL PREGNANT WOMEN TO PICK-UP WOMEN AT RISK FOR DEPRESSION AND I KNOW THAT THIS IS HAPPENING ALREADY FOR LONG TIME IN UK. IN OTHER WORDS, I DO NOT SEE THE ADDITIVE VALUE OF COMPUTER DIAGNOSTICS DURING PREGNANCY. OF COURSE THE AUTHORS ARE RIGHT THAT DURING THE POSTPARTUM PERIOD THE REGULAR CONSULTATIONS HAVE DISAPPEARED. THIS PERIOD SEEMS TO BE AN ATTRACTIVE PERIOD FOR COMPUTER BASED DIAGNOSTICS. HOWEVER, IN THE SUMMARY THEY STATE THAT THE CURRENT PROPOSAL ESPECIALLY FOCUS ON PREGNANCY.

With our research we are not proposing isolated computer diagnostic systems or to replace clinicians. The overall aim of the research is to explore the use of mobile technology to facilitate the implementation of clinical recommendations, in our case, NICE guidelines.

A previous feasibility study (now completed) assessed the use of tablets (and the potential impact of this delivery mode on data quality) in the waiting area of antenatal clinics in the NHS to implement NICE recommendations (i.e., Whooley Questions and EPDS). The results of which were communicated to the obstetric health care team to be used with the patient in the antenatal appointment.

The feasibility study presented in this manuscript proposes to explore the use of smartphones for the ongoing monitoring/screening of depression throughout pregnancy.

The aim is to use the data gathered outside the consultation to support a discussion between a pregnant woman and her obstetric healthcare worker during the consultation (our full list of potential advantages of a system like this are cited in the introduction).

However, patient adherence is a crucial issue with ongoing data collection (particularly in this case as we are dealing with self-reports), especially over a prolonged period of time (in this case 6 months). Hence, adherence to a sampling protocol and drop-out rates being the main outcomes of this feasibility study.

A FINAL COMMENT IS THE NUANCE THAT DEPRESSION IS A COMMON DISORDER DURING PREGNANCY AND POSTPARTUM. YES, THIS IS TRUE, BUT DEPRESSION IN THE PERINATAL PERIOD IS NO DIFFERENT IN PREVALENCE AND CHARACTERISTICS COMPARED TO NON-CHILDBEARING WOMEN OF THE SAME AGE. THIS GOES BACK TO LEWINSOHN'S WORK SHOWING THAT THE TWO MOST IMPORTANT DETERMINANTS OF DEPRESSION ARE: BEING YOUNG AND BEING FEMALE.

We have now added to the introduction that the prevalence and incidence of depression in pregnant women are not different from those in non-pregnant women. Nonetheless, research also suggests that the rate of recognition is low during pregnancy.

PAGE 4: STRENGTHS OF THE STUDY: "THIS STUDY EXPLORES THE ROLE OF MOBILE TECHNOLOGY AS A MEDIUM TO (I) ADDRESS SOME OF THE BARRIERS PREVENTING DEPRESSION SCREENING IN ANTENATAL SETTINGS", BARRIERS OF PATIENTS OR HEALTH CARE WORKERS??? SEE MY PREVIOUS COMMENTS.

We have now specified that we are referring to practical barriers facing both patients and healthcare workers. These are listed in the introduction.

SAME PAGE: "THIS STUDY WILL PROVIDE BASELINE INFORMATION REGARDING THE APPROPRIATENESS OF A SAMPLING PROTOCOL (IN TERMS OF ITS DURATION, INTENSITY AND FREQUENCY) FOR THE MONITORING OF MOOD AND DEPRESSION, I WOULD SAY: MOOD PROBLEMS.

We have now specified that this protocol is for the monitoring of mood and the screening of depression.

'ANTENATAL DEPRESSION IS ONE OF THE MOST COMMON, TREATABLE MENTAL HEALTH DISORDERS IN PREGNANCY.[1-4]'
UP TO 2% OF PREGNANT WOMEN ALREADY SUFFER FROM CHRONIC DEPRESSION IN WHICH TREATMENT FAILS TO SHOW LITTLE IF ANY BENEFIT.

Thank you for this information. We have removed the word treatable from this paragraph.

LINE 42 PAGE 5 TO LINE 9 OF PAGE 6: ALL THE ADVANTAGES OF SMARTPHONE USE CAN EASILY BE OVERRULED BY THE SIMPLE IMPLEMENTATION OF STANDARD MENTAL HEALTH DIAGNOSTICS DURING THE REGULAR OBSTETRIC CONSULTATION. MOREOVER, THE CLINICAL VIEW OF A MIDWIFE HAS TREMENDOUS ADVANTAGE OVER SELF-RATING REPORTS (BE IT BY SMARTPHONE OR PAPER AND PENCIL). THEREFORE I REALLY DOUBT THE COST-EFFECTIVENESS OF SMARTPHONE USE DURING PREGNANCY (NOT THE POSTPARTUM PERIOD).
DURING CONSULTATION THERE IS NO RISK OF RETROSPECTIVE INFORMATION: THE CLIENT SITS IN FRONT OF THE HEALTH CARE WORKER.

We are not proposing to use smartphones as a way of replacing clinicians. Instead, as per our previous answer, we are proposing the use of smartphones as a medium to gather clinical information, which can then be used in the obstetric consultation. Regarding your point of cost-effectiveness, there are issues around data management that can be addressed by the use of smartphones such as printing costs, data storage and data entry. In addition, there may be an added benefit for women of being able to provide them with information about variability in their mood and other domains.

Whilst in an individual consultation the healthcare worker can indeed contextualise the information, in UK practice mood is still primarily assessed just by questionnaire, and so the risks of retrospective information are still present if the midwife uses an instrument such as the EPDS (which is common practice in antenatal clinics in the UK) as this tool asks about the past 7 days. Questions about patients' wellbeing are also likely to be posed in a retrospective fashion (for example, How have you been feeling?).

PAGE 7: "RETROSPECTIVE ASSESSMENT: REQUIRING THE COMPLETION OF THE EPDS AT IRREGULAR INTERVALS ONCE A MONTH FOR 6 MONTHS," IRREGULAR INTERVALS ONCE A MONTH FOR 6 MONTHS SEEMS VERY REGULAR TO ME.

Thank you for highlighting this. We consider it to be an irregular interval because the alerts will be generated at random days within each month. Nonetheless, we have removed the phrase 'at irregular intervals' in this section [now page 8 in the revised manuscript].

PAGE 7-8: EXCLUSION CRITERIA: UP TO 10% OF THE PREGNANT WOMEN IN THE NETHERLANDS USE BENZODIAZEPINES / ANTIDEPRESSANT DRUGS. ARE THEY EXCLUDED IN THE CURRENT STUDY WHEN A WOMAN HAS NO DEPRESSION, WHAT MEANS THE WORD CURRENTLY BEING TREATED?

This includes any form of treatment, whether it is talking therapies or pharmacological treatment. We have added a sentence to the inclusion criteria section to clarify this.

SAME PAGE: "POTENTIAL PARTICIPANTS WILL HAVE AT LEAST 24 HOURS TO DECIDE ON PARTICIPATION" BEING A FORMER MEMBER OF A MEDICAL ETHICAL REVIEW BOARD, THE MINIMUM OF ONE WEEK WAS COMMONLY REQUESTED WHEN RECRUITING PARTICIPANTS.

This point was discussed during our meeting with the research ethics committee, and we obtained permission to use this approach. Participants will have a minimum of 24 hours (which is the minimum requirement of the sponsor), but in reality they can take as much as time as they wish. We have now rephrased this to indicate that participants will have 'a minimum of 24 hours'.

SAME PAGE: "WE WILL ADMINISTER AN 11-QUESTION SURVEY TO COLLECT INFORMATION ABOUT PARTICIPANTS' AGE GROUP, ETHNIC BACKGROUND, MARITAL STATUS, EMPLOYMENT STATUS, LEVEL OF EDUCATION, SMARTPHONE AND TABLET COMPUTER OWNERSHIP, OBSTETRIC HISTORY, AND PREVIOUS PERSONAL HISTORY OF DEPRESSION." PLEASE GIVE DETAILS: FOR EXAMPLE, UNPLANNED PREGNANCY IS AN IMPORTANT DETERMINANT OF PREGNANCY DEPRESSION. COMPLICATIONS DURING PREVIOUS PREGNANCIES / DELIVERIES.

The 11-question survey was included as in Appendix 1 in the original submission. I am afraid that we are not collecting information about all determinants (which has been acknowledged in the limitations of this feasibility study).

PAGE 9: " THE EPDS IS A 10-ITEM SELF-ADMINISTERED SURVEY THAT WAS DEVELOPED TO SCREEN FOR PERINATAL DEPRESSION IN THE COMMUNITY.[24]" NO, IT WAS ORIGINALLY DEVELOPED TO SCREEN FOR POSTPARTUM DEPRESSION.

We have now specified that the EPDS was originally developed to screen for postpartum depression and that since then it has been validated for use in the perinatal period and in community and clinical settings.

SAME PAGE: "SCORES OF 13 POINTS OR MORE SUGGEST THAT THE DIAGNOSTIC CRITERIA FOR MAJOR DEPRESSION DISORDER HAVE PROBABLY BEEN MET.[25]" OUR GROUP WAS AMONG THE FIRST TO VALIDATE THE EPDS DURING PREGNANCY. WITH ACCEPTABLE SENSITIVITY / SPECIFICITY, THE PPV OF A SCORE OF 13 OR HIGHER WILL NEVER BE HIGHER THAN 50-60%. WHAT DO THE AUTHORS MEAN BY "PROBABLY"? I AM AWARE OF A LARGE (THOUSANDS OF PARTICIPANTS) MULTI-CENTER META-ANALYSIS WHICH IS CURRENTLY UNDER WAY TO QUESTION THE RELIABILITY OF THE EPDS AS AN APPROPRIATE SCREENING INSTRUMENT FOR DEPRESSION IN THE PERINATAL PERIOD.

We have rephrased this section to indicate that we acknowledge the variability in the sensitivity, specificity, PPV and NPV of the EPDS cut-off scores depending on the setting and the population. With this feasibility study, we do not intend to question the reliability of the EPDS as an appropriate screening instrument; the main aim is to determine participants' adherence with the proposed sampling protocol.

SAME PAGE: "WE WILL ADMINISTER 5 MOMENTARY QUESTIONS TO ASSESS PARTICIPANTS' MOOD, SLEEP, WORRY, ENJOYMENT AND ENERGY (APPENDIX 3). THESE QUESTIONS ARE BASED ON THE WORK OF A RESEARCH FELLOW AT THE COLLABORATION FOR LEADERSHIP AND APPLIED HEALTH RESEARCH AND CARE (CLAHRC) FOR THE EAST OF ENGLAND.[26] EACH QUESTION WILL BE MAPPED ONTO 5-POINT PICTORIAL SCALES, RANGING FROM 1 (LOW) TO 5 POINTS (HIGH). FOR THE PURPOSE OF THIS FEASIBILITY STUDY, WE WILL NOT PERFORM ANY OVERALL SCORE CALCULATION OR ATTEMPT ANY VALIDATION OF THESE QUESTIONS." SO THEN, WHAT IS THE RELEVANCE OF ADDING THESE QUESTIONS IF IT IS NOT FOR VALIDATION? STRICTLY SPEAKING: WORRYING IS A MAJOR SYMPTOM OF ANXIETY WHICH IS NOT A MOOD SYMPTOM. ONCE AGAIN, THE LOW ENERGY QUESTION WILL NOT DISCRIMINATE AT ALL DURING PREGNANCY.

The relevance of adding these questions is to assess if they affect adherence to our proposed sampling protocol.

PAGE 12 " IN THIS EXPERIMENTAL MANIPULATION, PARTICIPANTS WILL BE ASKED TO COMPLETE A COMBINATION OF RETROSPECTIVE ASSESSMENTS AND MOMENTARY ASSESSMENTS. A RETROSPECTIVE ASSESSMENT WILL BE DEFINED AS A SINGLE ADMINISTRATION OF THE EPDS. A MOMENTARY ASSESSMENT WILL BE DEFINED AS THE 5 MOMENTARY QUESTIONS PLUS THE 2 CONTEXTUAL QUESTIONS." TO ME THIS IS A MAJOR FLAW OF THE DESIGN: THE AUTHORS USE A NON-VALIDATED INSTRUMENT (5 MOMENTARY QUESTIONS) TO DISCRIMINATE BETWEEN TWO ARMS OF A PROTOCOL.

The main aim of this feasibility study is to assess patient engagement (in the form of adherence to a sampling protocol and drop-out rates). One of the main issues with engagement is the tension between clinicians' informational needs and the value that patients derive from these measures. The addition of the momentary questions was an attempt to help participants make sense of their experience whilst collecting clinical data (since the domains chosen for this questions were derived from qualitative work in which women were asked what areas they would find useful to track), and hence improve engagement.

SAME PAGE: "THE ASSESSMENT PERIOD WILL BE STRUCTURED AS FOLLOWS:
DAY 1: ONE RETROSPECTIVE ASSESSMENT AT ANY RANDOM TIME BETWEEN 17:00 - 21:00;
DAY 2 TO 5: 3 MOMENTARY ASSESSMENTS PER DAY, DISPLAYED AT RANDOM TIMES WITHIN

EACH OF THE FOLLOWING INTERVALS: 09:00 – 12:00; 13:00 – 16:00; AND 17:00 – 20:00; AND DAY 6: ONE RETROSPECTIVE ASSESSMENT AT ANY RANDOM TIME BETWEEN 17:00 AND 21:00.”
THE LATTER RETROSPECTIVE REFERS TO THE PREVIOUS 7 DAYS?

It does.

PAGE 15: “WE HAVE CHOSEN TO RELATE THE PROPOSED SAMPLE SIZE TO THE 95% CONFIDENCE INTERVAL FOR THE ADHERENCE RATE, AS RECOMMENDED BY THE RESEARCH DESIGN SERVICE IN LONDON. THEREFORE, WE WOULD NEED 96 PARTICIPANTS IN EACH ARM WITH A 95% CONFIDENCE LEVEL AND A CONFIDENCE INTERVAL OF 10. THIS TRANSLATES INTO A TOTAL SAMPLE OF 192 (I.E., 96 PARTICIPANTS IN EACH EXPERIMENTAL GROUP). FOR THIS REASON, WE WILL AIM TO RECRUIT AT LEAST 200 PARTICIPANTS, TO ACCOUNT FOR DROP-OUTS.”
BECAUSE THE AUTHORS START VERY EARLY DURING PREGNANCY DO THEY INCORPORATE ABORTION DROP-OUT, WHICH CAN BE SUBSTANTIALLY UNTIL 16 WEEKS?

We have now amended this section to indicate that we will aim to recruit 250 participants to account for drop outs and potential miscarriages.

SAME PAGE: “WE WILL COMPARE THE RETROSPECTIVE AND MOMENTARY ASSESSMENT AND THE RETROSPECTIVE ASSESSMENT EXPERIMENTAL GROUPS FOR DIFFERENCES IN ADHERENCE RATES, DROP-OUT RATES AND TIMELINESS OF DATA COMPLETION.” TO BE HONEST, UNTIL THIS PARAGRAPH, THE RATIONALE FOR THE COMPARISON OF THESE TWO APPROACHES WAS NOT CLEAR TO ME AND STILL IS NOT VERY CLEAR. NORMALLY IN TRIALS, THERE IS A (NULL) HYPOTHESIS: WHAT DO THE AUTHORS EXPECT AND BASED ON WHICH ASSUMPTIONS?

We have decided to go for a non-directional hypothesis (i.e., assessing the presence of differences between the 2 groups). One could argue that since the retrospective plus momentary assessment represents an increased burden to respondents (due to a longer duration of the assessment and more questions), that there will be lower adherence and higher drop-out rates in this group. Alternatively, one could argue that the EPDS is a clinical tool, which has limited meaning for many pregnant women, therefore the inclusion of momentary questions dealing with domains they can relate to could improve adherence and reduce drop-out rates. These phenomena have been reported in the literature.

We have reworded some sections at the end of the introduction to make the rationale for this comparison more explicit.

PAGE 53: THE RATIONALE OF THE CONTEXTUAL QUESTIONS IS NOT CLEAR TO ME. WHAT DO THE AUTHORS THINK TO CONCLUDE FROM THE ANSWERS GIVEN BY THE RESPONDENTS?

Evidence suggests that answers given to momentary questions can be influenced by the context in which they are completed. As part of this feasibility study, we want to conduct an exploratory analysis to assess if this is the case.

VERSION 2 – REVIEW

REVIEWER	Prof. Dr. VJ Pop, MD, PhD Department of Clinical and Medical Psychology Tilburg University The Netherlands
REVIEW RETURNED	23-Dec-2016

GENERAL COMMENTS	The authors have responded adequately to my comments on the first submission.
---